# Proof-of-Concept for Liquid Biopsy Disease Monitoring of *MYC*-Amplified Group 3 Medulloblastoma by Droplet Digital PCR

**DOI:** 10.3390/cancers15092525

**Published:** 2023-04-28

**Authors:** Natalia Stepien, Daniel Senfter, Julia Furtner, Christine Haberler, Christian Dorfer, Thomas Czech, Daniela Lötsch-Gojo, Lisa Mayr, Cora Hedrich, Alicia Baumgartner, Maria Aliotti-Lippolis, Hannah Schned, Johannes Holler, Katharina Bruckner, Irene Slavc, Amedeo A. Azizi, Andreas Peyrl, Leonhard Müllauer, Sibylle Madlener, Johannes Gojo

**Affiliations:** 1Department of Pediatrics and Adolescent Medicine, Comprehensive Center for Pediatrics and Comprehensive Cancer Center, Medical University of Vienna, 1090 Vienna, Austria; natalia.stepien@meduniwien.ac.at (N.S.); sibylle.madlener@meduniwien.ac.at (S.M.); 2Division of Neuroradiology and Musculoskeletal Radiology, Department of Biomedical Imaging and Image-Guided Therapy, Medical University of Vienna, 1090 Vienna, Austria; 3Research Center for Medical Image Analysis and Artificial Intelligence (MIAAI), Faculty of Medicine and Dentistry, Danube Private University, 3500 Krems-Stein, Austria; 4Division of Neuropathology and Neurochemistry, Department of Neurology, Medical University of Vienna, 1090 Vienna, Austria; 5Department of Neurosurgery, Medical University of Vienna, 1090 Vienna, Austria; 6Department of Pathology, Medical University of Vienna, 1090 Vienna, Austria

**Keywords:** liquid biopsy, droplet digital PCR, medulloblastoma, group 3, *MYC* amplification, minimal residual disease, therapy monitoring, cerebrospinal fluid, pediatric oncology

## Abstract

**Simple Summary:**

*MYC* is a well-described oncogene across multiple cancer types. In medulloblastoma (MB), significance is subtype-dependent and associated with a particularly dismal outcome in *MYC*-amplified group 3 MBs. In our study, we established a highly sensitive and specific method for the detection of *MYC* amplification in cerebrospinal fluid (CSF) enabling its use as a liquid biopsy biomarker. We show preliminary results of its potential as a marker for early diagnosis, disease staging and monitoring. The fast and cost-effective method could substantially improve means of diagnosis and therapy monitoring in high-risk MBs.

**Abstract:**

Background: Liquid biopsy diagnostic methods are an emerging complementary tool to imaging and pathology techniques across various cancer types. However, there is still no established method for the detection of molecular alterations and disease monitoring in MB, the most common malignant CNS tumor in the pediatric population. In the presented study, we investigated droplet digital polymerase chain reaction (ddPCR) as a highly sensitive method for the detection of *MYC* amplification in bodily fluids of group 3 MB patients. Methods: We identified a cohort of five *MYC*-amplified MBs by methylation array and FISH. Predesigned and wet-lab validated probes for ddPCR were used to establish the detection method and were validated in two *MYC*-amplified MB cell lines as well as tumor tissue of the *MYC*-amplified cohort. Finally, a total of 49 longitudinal CSF samples were analyzed at multiple timepoints during the course of the disease. Results: Detection of *MYC* amplification by ddPCR in CSF showed a sensitivity and specificity of 90% and 100%, respectively. We observed a steep increase in amplification rate (AR) at disease progression in 3/5 cases. ddPCR was proven to be more sensitive than cytology for the detection of residual disease. In contrast to CSF, *MYC* amplification was not detectable by ddPCR in blood samples. Conclusions: ddPCR proves to be a sensitive and specific method for the detection of *MYC* amplification in the CSF of MB patients. These results warrant implementation of liquid biopsy in future prospective clinical trials to validate the potential for improved diagnosis, disease staging and monitoring.

## 1. Introduction

Brain tumors are the most common solid malignancies in the pediatric population and the leading cause of cancer-associated death due to the high aggressiveness of certain tumor types [1]. Although diagnostic methods and therapeutic possibilities have constantly advanced over the last decades, some tumors still lack curative treatment options and diagnosis remains challenging [2]. Research has thus focused on characterizing molecular markers for diagnosis, prognosis, and prediction by analyzing tumor tissue, mainly at the time of diagnosis [3,4]. However, there is still a pressing need for disease monitoring tools, methods to better depict tumor evolution and additional techniques for diagnosis of radiologically inconclusive cases.

The *MYC* oncogene with its two paralogues *MYCN* and *MYCL* is one of the most important oncogenes across many human cancer types, including a subgroup of pediatric CNS tumors [5,6,7]. It is a master regulator of multiple cellular programs and is therefore a key factor for the development of representative preclinical cancer models. Its diagnostic, prognostic and predictive significance depends strongly on the tumor entity [4,5,6]. In the case of MB, the most common malignant CNS neoplasm in the pediatric population, its clinical significance even differs depending on subgroup. Despite high expression of MYC in WNT MB, the prognosis is excellent, whereas amplification of *MYC* in metastatic group 3 tumors is an additional negative prognostic factor [8]. Moreover, *MYCN* amplification is most commonly found in SHH and group 4 MBs [3]. Currently, the presence of *MYC/N* amplification is decisive for treatment stratification when following the SIOPE PNET 5 MB protocol [9]. Therefore, rapid molecular diagnosis may lead to improved clinical care by enabling the treating physicians to specify required therapy at an earlier timepoint. Group 3 MBs tend to recur early and in a metastatic pattern with poor survival after recurrence [10]. Novel treatment strategies for recurrent MBs have recently been published [11] or are currently under investigation (NCT01356290; NCT04743661; NCT04501718). Early detection of recurrence and dense response monitoring will be an essential component in the implementation of newly discovered treatment options for this patient collective with a dismal prognosis.

Therapeutic planning and response monitoring of *MYC*-amplified MB are currently based on imaging techniques, which are necessary to monitor tumor size and most importantly provide information about anatomical location and metastases. However, magnetic resonance imaging (MRI) is time consuming and frequently requires anesthesia in young children, limiting the time resolution in a very dynamic disease. Moreover, MRI provides limited insight into spatial resolution and may miss “minimal residual disease” or microscopic tumor residues [12]. Lastly, despite constantly improving imaging techniques, in some cases tumor tissue cannot unambiguously be distinguished from non-malignant tissues or therapy-associated image changes [13]. Consequently, more accurate and easily accessible monitoring techniques are urgently needed. Liquid biopsy (LB) addresses the above-mentioned limitations by adding insight into molecular patterns and biological tumor information, thereby depicting tumor heterogeneity and providing a non-invasive strategy to monitor the molecular development of CNS malignancies in response to therapy. Liu et al. recently revealed how LBs of MB patients analyzed by low-coverage whole genome sequencing (lc-WGS) enabled early detection of molecular alterations within primary treatment, which were later confirmed in tissues of metastases or recurrent tumors [14]. Building on these findings, LB may also provide molecular information at a time of metastatic recurrence, when tissue biopsy is not feasible anymore, thereby facilitating the use of targeted therapies in a 2nd or 3rd line setting.

LB performed at each MRI timepoint may aid in the detection of minimally residual disease, thereby allowing for continuous adaptation of individualized maintenance chemotherapy. Furthermore, it could potentially improve informed decision-making when it comes to therapeutic planning in cases with unclear MRI results. Generally, regarding CNS tumors, current techniques for LB perform better in CSF samples than in blood samples [15]; however, acquisition of CSF is regarded to be more invasive than drawing blood. Addressing this issue, our research group has published several articles on the use of Ommaya reservoirs for frequent intraventricular therapy, mainly in the recurrent MB setting, but also for other treatment protocols requiring repeated intraventricular therapy administration [16,17]. The procedure is minimally invasive, regarded by the patients as less painful than a blood draw and, if performed cautiously [16], associated with minimal risks [17]. Currently, intraventricular therapy is recommended for young children as a radiation-sparing option in the present HIT-MED Guidance protocol for patients with newly diagnosed medulloblastoma [18,19] or similar protocols for this age group. Moreover, intraventricular therapy is a means of therapy intensification in the recurrent setting included for example in the MEMMAT protocol [11]. Therefore, an Ommaya reservoir is frequently already in place and the performance of liquid biopsies upon application of therapy is associated with no additional risk. Considering the detrimental biological behavior of recurrent *MYC*-amplified MB and the persisting lack of standardized therapy upon progression or relapse, repeated assessment of disease burden as well as therapeutic response is essential in improving patient care and survival. LB, performed on CSF obtained via an Ommaya reservoir, may facilitate disease monitoring by providing rapidly available results, thereby enabling continuous treatment adaptation, depending on the patient’s individual response.

We and others have already demonstrated the applicability of LB in various pediatric brain tumor types [14,20,21,22], including our study on the monitoring of disease burden by *NRAS* detection in CSF samples of a patient with primary diffuse leptomeningeal melanomatosis [2]. Within this study, we found that the defining molecular alteration was better-detectable in CSF than in the biopsy tissue, highlighting the sensitivity of this method. Therefore, LB might also aid in the timely diagnosis of unspecific CNS lesions.

In the present study, we selected ddPCR as the detection method of *MYC* amplification, utilizing its extremely high sensitivity for singular molecular alterations. We show that this molecular alteration can be tracked in the CSF of MB patients. Importantly, liquid biopsy detection of *MYC* amplification harbors potential in disease staging and monitoring as well as early diagnosis as further discussed in our representative case.

## 2. Materials and Methods

### 2.1. Human Tissue/Body Fluids

This study was conducted according to the principles expressed in the Declaration of Helsinki. The study protocol was approved by the Ethics Committee of the Medical University of Vienna (EK 1244/2016), and written informed consent was obtained from all study participants and/or their parents. Patients included in the study were being treated for a CNS malignancy at the Department of Pediatrics and Adolescent Medicine, Medical University of Vienna. Patient characteristics are provided in Appendix A. Results of methylation arrays performed on tumor tissue from patients with MB were screened for the presence of *MYC* amplification. The methylation arrays used were the Infinium Human Methylation 450K BeadChip (Illumina, Isc; San Diego, CA, USA) and the Infinium Methylation EPIC Array (Illumina, Isc; San Diego, CA, USA) (Appendix A). An integrated diagnosis was used to define MB group 3 patients, and CNV plots from the methylation analysis were used to determine *MYC* amplification level (Appendix A). We identified 5 cases of group 3 non-WNT/non-SHH MB (4 subgroup II, 1 subgroup III) [23]. Log2 copy number ratios (CNR) estimated from the CNV plots were used to calculate an estimated AR as given in Table 1 using the following formula: (2^CNR^)*2. CSF samples were collected by the puncture of an Ommaya reservoir only with clinical indication, mostly directly before the application of intraventricular chemotherapy. Three CSF samples were obtained by lumbar puncture (LP), only with clinical indication (Pat 1, sample 1 and 2; Pat 5, sample 2). One CSF sample was obtained intraoperatively (Pat 2, sample 1) and two samples were obtained from an external ventricular drain (EVD, Pat 5, sample 1 and 2). CSF samples were frozen immediately after collection at −20 °C and later on transferred to long-term storage at −80 °C. Blood samples were obtained with clinical indication and directly centrifuged for 10 min at 3000 rpm (=1.4 rcf); plasma/serum was then transferred into a collection tube and stored at −80 °C. Tissue samples were obtained at the time of tumor surgery, frozen immediately after collection and stored at −80 °C. All samples were collected between 2014 and 2023. Analysis was performed in the years 2021–2023, retrospectively for patient 1–4 and during the course of disease for patient 5. Selection of samples for analysis was performed considering clinically important events, dates of MRI and availability of material. Additionally, ten samples acquired on five consecutive days from two patients were analyzed for clinical validation. Methods and patient samples are summarized in Appendix A.

### 2.2. Clinical and Radiological Data

Clinical and radiological data of the patients were derived from multiple sources of information, including print and digital patient charts. Additionally, all relevant MR images were reviewed at the time of treatment within an interdisciplinary CNS tumor board, including radiologists, surgeons, radio- and neuro-oncologists and treating physicians as well as retrospectively by an independent radiologist at the time of sample analysis. CSF cytology analysis was performed routinely at each sampling timepoint for monitoring purposes. Results were reported as either unsatisfactory, negative, abnormal, suspicious or malignant. For purposes of analysis, cytology examinations reported as “suspicious” were scored as positive, and those interpreted as “abnormal” were scored as negative.

### 2.3. Medulloblastoma Cell Lines

For method validation D425Med and D341Med cells were obtained from American Type Culture Collection (ATCC, Manassas, VA, USA). Both human medulloblastoma cell lines are classified as group 3 MB cell lines with *MYC* amplification.

### 2.4. DNA Isolation

The “Quick-cfDNA/RNA serum & plasma kit” from Zymo Research (Zymo Research Corp., Irvine, CA, USA) was used for cell free/circulating tumor (cf/ct) DNA isolation. All samples were centrifuged to remove cell debris. The isolation of cf/ctDNA was performed according to the manufacturer’s instructions, except for one minor modification, that is, the double elution with 20μL using the same column in order to increase cfDNA concentration. Amounts of isolated DNA were measured by the nanodrop 1000 (Thermo Fisher Scientific, Waltham, MA, USA). For tissue DNA isolation the ReliaPrepTM gDNA Tissue Miniprep System was used (Promega, Madison, WI, USA).

### 2.5. Droplet Digital PCR

Predesigned and wet-lab validated ddPCR probes for *MYC* were used for detection of *MYC* amplification (Assay ID: *MYC*-probe: dHsaCP2506523; AP3B1 (control) probe: dHsaCP2500315). ddPCR was performed in duplicates on 2 × 5 μL isolated DNA using the “Droplet Digital PCR System” from BioRad (Bio-Rad Laboratories Inc., Hercules, CA, USA) according to the manufacturer’s instructions. Amplification rate (AR) was calculated using the following formula: AR = [(copies of gene of interest/copies of reference gene)] × 2.

### 2.6. Data Analysis and Visualization

Droplets were analyzed using QuantaSoft Version 1.7.4.0917 (Bio-Rad Laboratories Inc., Hercules, CA, USA). Data analysis was performed using GraphPad Prism 8.0.1 (GraphPad Software Inc., San Diego, CA, USA). The graphical abstract was created with BioRender.com. The swimmer plot was created using Pages, version 12.2.1 (Apple Inc., CA, USA) and Paint 3D (Microsoft Corp., Sacramento, CA, USA).

## 3. Results

In a first step, a reliable and rapid protocol for detection of *MYC* in bodily fluids was established. To confirm the detectability of *MYC* by our probes, two MB cell lines with known *MYC* amplification were used. After DNA isolation, three different concentrations were used to evaluate the sensitivity of ddPCR for *MYC* detection. *MYC* amplification was detected at all concentration levels in both cell lines, with the lowest concentration used being 0.001 ng. Variability between experiments performed in duplicate decreased with increasing DNA amount (Figure 1). The median *MYC* AR was 102.6 and 58.9 for D425 and D341 cells, respectively, overall confirming the applicability of the method to detect minimal amount of *MYC*-amplified DNA.

FISH analysis to confirm our methylation array-based classification of the presence of *MYC* amplification was available for 4/5 patients (patients 2–5). FISH analysis of the tumor tissue from patient 4 showed a gain of *MYC* in 63.5% of cells, but no amplification as per definition (ratio < 4) (Table 1). We validated the newly established method on tumor tissue from tumors with *MYC* amplification (*n* = 3, pat 1, 3 & 5) and *MYC* gain (*n* = 1, pat 4). ARs were 17.2 (pat 1), 69.9 (pat 3), 3.3 (pat 4) and 39.2 (pat 5), which compares well to the descriptive evaluation of *MYC* amplification in tumor tissue as shown in Table 1. One tissue sample from a group-3/4-MB patient with *MYC* polysomy was used as a control, showing no significant *MYC* amplification as per ddPCR. Additionally, tissue from three non-MB, non-*MYC*-amplified tumors was used as negative controls, resulting in a median *MYC* AR of control tissue of 1.9.

Building on these promising results in tumor tissue samples, DNA was isolated from CSF samples of five MB patients with *MYC* amplification (*n* = 49). The average storage time at −80 °C was 7, 4, 8, 5 and <1 years for patients 1–5, respectively. The respective median *MYC* AR rate from CSF was 9.7, 3.2, 34.8, 1.6 and 25.9, thus not indicating an impact of storage time on AR in this limited dataset. AR of tumor tissue assessed by FISH or panel sequencing correlated well with tissue ARs determined by ddPCR as well as median AR in CSF (Table 1).

CSF samples from patients with non-*MYC*-amplified MB (*n* = 3), ependymoma (*n* = 1), low grade glioma (*n* = 1), craniopharyngioma (*n* = 1) and a patient with a suspected pituitary apoplexy without malignancy (*n* = 1) were used as negative controls and revealed values of 1.1; 1.7; 0.6; 1.8; 1.5; 1.6; and 2.1, respectively (Appendix A). Using a cut-off level of 2.2 for *MYC* amplification, as suggested by ROC analysis, ddPCR has a sensitivity of 90% and a specificity of 100% for detection of *MYC* amplification in CSF (Figure 2).

To further explore the potential of ddPCR for quantification of *MYC* amplification, and thereby that of therapeutic response monitoring, we performed two serial analyses of two groups of five consecutive samples acquired within one week from patient 1 and patient 3, respectively. This resulted in a mean *MYC* AR of 6.9 (SD: 1.6) for patient 1 and a mean of 25.3 (SD: 10.2) for patient 3 (Figure 3 and Appendix A). It is worth noting that both patients received intraventricular therapy with etoposide after CSF collection via the Ommaya reservoir on each of the days analyzed. Both treatment weeks were during regular MEMMAT cycles, including intraventricular therapy, with preceding and following treatment weeks.

To evaluate the detectability of *MYC* amplification in blood samples, nine matched blood samples from patients 1–5 were selected. Samples were selected by temporal proximity to a CSF sample analyzed, resulting in an interval of 0–31 days between positive CSF and investigated paired blood sample. Except one, all blood samples were negative. The positive sample showed an *MYC* AR of 2.7 and matched the extraordinarily high *MYC* AR result from CSF (AR 3247.9) in patient 5 (Figure 4).

Finally, we investigated the potential of *MYC* detection as a possible biomarker for monitoring MB disease. Serial liquid biopsies at clinically relevant timepoints, such as change of therapy or temporal proximity to MRI imaging of patients with *MYC*-amplified MB were used to investigate whether ddPCR shows potential for effective disease monitoring. Figure 5 depicts a swimmer plot indicating ARs (upper red line) and clinical events (lower blue bar) for each patient individually across the course of disease. The consecutive sampling described earlier is also outlined in the graphs, thereby qualifying the high variation found in the measurements of consecutive days depicted in Figure 3 and Appendix A. Molecular results were compared to MRI data and clinical events, depicted within the swimmer plot bars (Figure 5). Importantly, due to the aggressive nature of the disease, there was no CSF sample available for any patient off treatment or in complete remission. A proportion of 4/5 patients suffered from leptomeningeal dissemination already at first LB and were radiologically not tumor-free at any time point. Patient 5 was radiologically tumor-free after first surgery; nevertheless, LB was still positive two weeks after surgery. No CSF samples were available of this patient until radiological progression, including leptomeningeal dissemination and implantation of an Ommaya reservoir 9 months after diagnosis. The leptomeningeal disease present in 4/5 cases and in patient 5 at recurrence implicates a constant adjacency to the CSF. Representative images of all patients at diagnosis are included in Appendix A.

To rule out the influence of sampling site on the AR variability, we evaluated *MYC* AR in CSF derived intraoperatively/via LP or from an external ventricular drain (EVD). We further compared the results to the most temporally proximate AR from an Ommaya reservoir, as well as to the median Ommaya-derived AR from each patient. The results are depicted in Figure 6. It is worth noting that the samples were taken up to 10 months apart from each other (pat 5), as we also highlighted in Figure 5, thus limiting their comparability. *MYC* amplification was detectable at all sample sites, including the EVD sample, which is remarkable since CSF was stored over a period of several hours, without any cooling, within the drainage system. Considering our limited data set, there is no evidence of superiority of one sampling site in the setting of metastatic *MYC*-amplified MB.

Cytology showed tumor cells in both LPs but not for the 1st Ommaya puncture after the 2nd LP in patient 1. In patient 2 it was only positive at the day of surgery, and negative at the 1st Ommaya puncture. In patient 5 cytology was positive at the day of EVD sampling, as well as on day 14, but negative at 1st Ommaya sampling. To further compare ddPCR to cytology, we analyzed the cytological results from all CSF collections performed as part of routine disease monitoring. Patients 1–5 showed a positive cytology in 5.4%, 16.8%, 16.3%, 17.3% and 11.4% of all cytologically inspected samples, respectively. Only at one timepoint ddPCR detection was negative while cytology was positive. This was the case in patient 4 who harbored only an *MYC* gain and generally rather low AR (Figure 5). Vice versa, cytology was negative in 81.6% of 38 matched LB-positive samples, suggesting a higher sensitivity of ddPCR as compared to cytology. Notably, cytology was frequently alternating between positive and negative results within the same treatment period. Therefore, considering the permanent radiologically visible tumor load, LB provides a more consistent image of disease activity when compared to CSF cytology.

Lastly, we performed an in-detail review of the clinical history of patient 2. An eight-year-old female presented to a routine ophthalmological exam where reduced vision was noted prompting an MRI (Appendix A, image 2A). The MRI showed thickening of the optic nerve with contrast enhancement, which led to a suspected diagnosis of optic neuritis, leading to the initiation of steroid treatment. Only upon repeated MRI (Appendix A, image 2B) was a mass in the 4th ventricle and further contrast enhancement in the cerebellar folia revealed, leading to biopsy and eventually diagnosis of disseminated MB. The fact that we detected *MYC* amplification in the CSF in our retrospective analysis suggests that a tumor could have been suspected already upon analysis of CSF, via lumbar puncture performed as diagnostic workup upon suspicion of optic neuritis. Taking these findings into consideration, future CSF workup of suspicious CNS lesions could also include tumor copy number variations, thus accelerating diagnosis in diagnostically challenging cases.

## 4. Discussion

Screening for minimal residual disease by molecular techniques is a mainstay in the longitudinal monitoring of hematological diseases and has recently been suggested for MB as well [14,24]. However, rapid and robust techniques for detection of high-risk markers from liquid biopsies are still lacking.

One of the biggest advantages of ddPCR is the exceptionally high sensitivity for detection of singular molecular alterations [25,26]. In most pediatric CNS tumor patients only small amounts of material are available for LB. In our case series, important molecular information could be retrieved from only 0.5–1 mL of CSF, corresponding to 20μL isolated cfDNA and a cfDNA concentration of <10 ng/μL [2]. All samples were collected during routine therapeutic or diagnostic procedures and accessing of Ommaya reservoirs was performed by multiple treating physicians. Considering the rapid development of LB techniques in recent years and the ongoing implementation into clinical practice, development of a robust and safe method with minimal pre-analytical requirements is of utmost importance to ensure regular sampling. In the presented study, the freshly acquired CSF was aliquoted into 2 mL cryogenic vials and stored at −20 °C without primary centrifugation. Samples were then transferred every 2–4 months into a −80 °C freezer. This pre-freezing step allows for immediate freezing by the treating physician without the need for special equipment or additional processing time, thereby not disrupting routine clinical care. Furthermore, we decided against the use of pre-defined augmented storage vials for DNA or miRNA preservation in order to minimize the pre-analytical complexity and allow for comparability with our retrospective cohort. The samples used in this study were finally analyzed up to eight years after collection, still resulting in very high sensitivity and specificity. The lack of correlation between storage time and detectability demonstrates the robustness of ddPCR proving its feasibility in routine clinical care. Additionally, it is a proof of concept that certain molecular alterations, including amplifications, can be easily detected in the CSF by ddPCR.

With respect to the potential for disease monitoring we found a mixed picture and no clear correlation to disease burden in our limited cohort. In patients 1 and 3 the *MYC* AR was constantly high, corresponding to their high disease burden. In patient 2, *MYC* amplification was constantly detected at all but a single time-point; shortly after treatment had been switched to antiangiogenic therapy including intraventricular therapy. However, the patient progressed shortly thereafter and finally succumbed to her disease. Therefore, it remains unclear whether the decrease in AR was due to short-time response to therapy or failure in detection. Patient 4 had low *MYC* amplification levels in the tumor material of a resected metastasis (first available methylation array, Appendix A); however, there was increased *MYC* amplification in the autopsy material as assessed by methylation analysis. FISH analysis finally revealed a gain of *MYC* as defined by a ratio below 4. The LB results showed correspondingly low overall ARs (<2.2.), thus were considered negative for *MYC* amplification. The sole CSF sample positive for *MYC* amplification (AR 7.7.) was collected 5 days after the resection of the metastasis, for which methylation analysis showed slightly increased *MYC* and FISH analysis displayed a gain of *MYC,* maybe indicating ctDNA following the surgical intervention. Patient number 5 had very high ARs in CSF at recurrence that decreased after a change of treatment modality to antiangiogenic therapy with intraventricular therapy coinciding with clinical stabilization. *MYC* amplification in CSF slowly decreased, possibly due to CSF clearance following intraventricular therapy. However, in the further course of disease MRI showed disease progression leading to the cessation of intraventricular therapy. A few months later, the LB showed a dramatic increase in *MYC* amplification and MRI confirmed massive tumor progression, while the patient’s clinical condition deteriorated. At this time the only positive blood sample was collected. As such, the results of our 5th patient are promising regarding the use of ddPCR for therapeutic response monitoring. In addition, patient 1, 2 and especially 5 showed an increase in AR in the terminal state.

However, there was a considerable variability in the AR detected by ddPCR, even in samples taken consecutively in the very short time frame of a single week (Figure 3 and Appendix A). This may on the one hand result from the increasing variability with decreasing cfDNA concentration (Figure 1), but on the other hand may also reflect natural variability of ctDNA shedding into CSF from tumor tissue, fluctuating CSF circulation (in particular in patients with VP Shunts) or response of tumor tissue to ongoing intraventricular therapy. In addition, variable preprocessing may also influence the measured AR. Based on this variability, measuring *MYC* amplification by ddPCR in CSF may not be suitable for short-term therapy monitoring, and the importance of ddPCR as a method for LB may rather be the qualitative detection of *MYC* amplification. Our preliminary data suggest a certain correlation of CSF AR detected by ddPCR with the AR as determined by FISH and other methods (Table 1); however, the applicability of the method for disease monitoring needs further longitudinal investigation in extended patient cohorts. Unfortunately, all our analyzed patients suffered from continuous tumor burden during the time span where CSF samples were collected, reflecting the deleterious nature of *MYC*-amplified MB. Further studies are needed to investigate the capability of ddPCR to detect *MYC* amplification or its absence in patients during complete remission. However, compared to cytology, LB was positive in a substantial proportion of cytologically negative samples, proving its superiority in detecting CSF positivity, especially given the clinical course of the patients. This should prompt a discussion on the evaluation of LB for staging purposes in addition to cytology on day 14 post surgery in upcoming clinical studies.

The study demonstrates the high sensitivity and specificity of *MYC* amplification detection in CSF, which allows for early diagnosis and more precise risk stratification. Using a cut-off level of two based on cell diploidy would include low-level amplifications or few unspecific outliers. Our current limited data suggest a cut-off level of 2.2 for detection of *MYC* amplification resulting in a sensitivity and specificity of 90% and 100%, respectively, which needs further evaluation within prospective clinical trials and extended patient cohorts. LB and ddPCR could be easily applied in patients who are primarily managed by a third ventriculostomy or the placement of an external ventricular drainage. Performance of LB on CSF obtained during third ventriculostomy or via a temporary ventricular drain would facilitate appropriate risk-stratified surgical management, i.e., aggressiveness of resection.

The rapid turn-around time of only six hours allows for real-time diagnostics, thereby answering clinically pressing questions much faster than conventional methods. The importance was emphasized by the case of patient 2, who was initially treated for optic neuritis. Similar to cases discussed in the literature, repeatedly there are patients presenting with unclear lesions on MRI, where surgery is associated with significant risks, who would benefit from LB to rapidly provide more insight into differential diagnoses [2,27]. To address tumor heterogeneity, performance of multiplex ddPCR assays with pre-specified tumor-specific probes warrant further investigation to become a new approach for the diagnosis of unspecific, tumor-suspicious CNS lesions. This is possible, since wet-lab validated probes for many singular molecular alterations are already commercially available and have been successfully used for ddPCR of CSF-derived ctDNA [2,28], and customized probes for patient-specific alterations are easily obtainable if needed for therapeutic response monitoring [26,29,30]. However, the main disadvantage of ddPCR in this application is the selectivity of analyzed genomic alterations. Each alteration of interest requires a specific probe, thereby increasing costs and the amount of material needed. Therefore, other methods such as whole exome sequencing or targeted sequencing may be preferred for specific questions [14,31,32].

Despite our promising results for *MYC* AR detection in CSF, reliable detection of *MYC* amplification in blood samples has proved impossible. There was one sample just above the cut-off level matching a CSF sample showing an extremely high AR of over 3000. Therefore, the possibility of detection of *MYC* amplification in blood needs further evaluation, and the present evidence clearly demonstrates the superiority of CSF for the detection and quantification of *MYC* amplification. The optimal source of biofluids for CNS tumor monitoring is still under discussion [20,25]. Keeping in mind the invasiveness of obtaining CSF, compared to blood or urine, the primary goal of LB development would be diagnosis and disease monitoring from urine, or secondary blood. However, many groups working on CNS tumors postulate a better detectability of CNS tumor signatures in the CSF, most likely due to the limited permeability of the blood brain barrier, the proximity of the tumor to the CSF compartment and the almost-DNA-free healthy CSF leading to a high percentage of tumor-derived DNA in the case of malignancy [21,29,33,34,35]. Considering the frequent need of Ommaya reservoirs in patients with high-grade CNS tumors such as MB, where the need for monitoring tools is exceptionally high, LB of CSF is an excellent option for these patients and further studies regarding the applicability of ddPCR for response monitoring are warranted.

## 5. Conclusions

Our study introduces ddPCR as a highly sensitive and specific method for detection of *MYC* amplification in the CSF of MB patients. Our work proves the feasibility of regular CSF collection via Ommaya Reservoirs and highlights the benefits of ddPCR. However, further studies exploring its applicability for detection of minimal-residual disease and therapeutic response monitoring are needed.

## Figures and Tables

**Figure 1 cancers-15-02525-f001:**
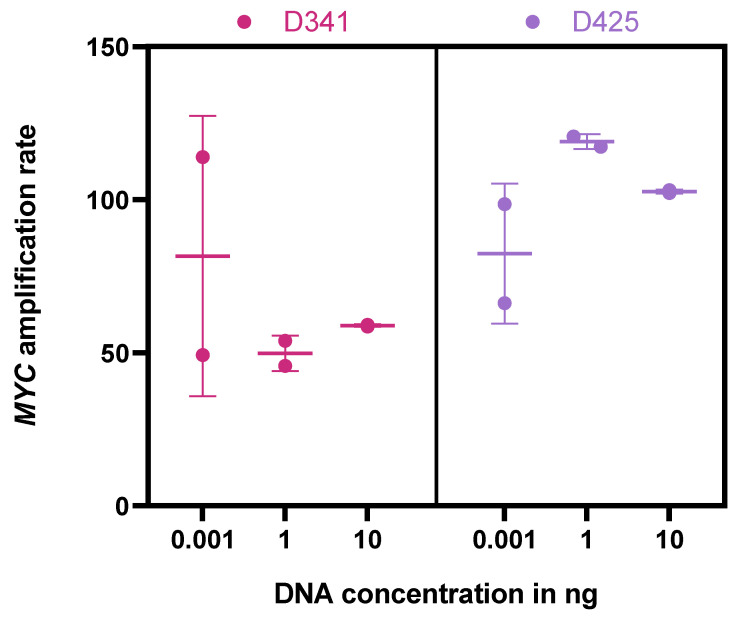
*MYC* AR by ddPCR in D341 and D425 cells. Isolated DNA diluted to specific concentrations (0.001 ng, 1 ng, 10 ng). Experiments were performed in duplicates.

**Figure 2 cancers-15-02525-f002:**
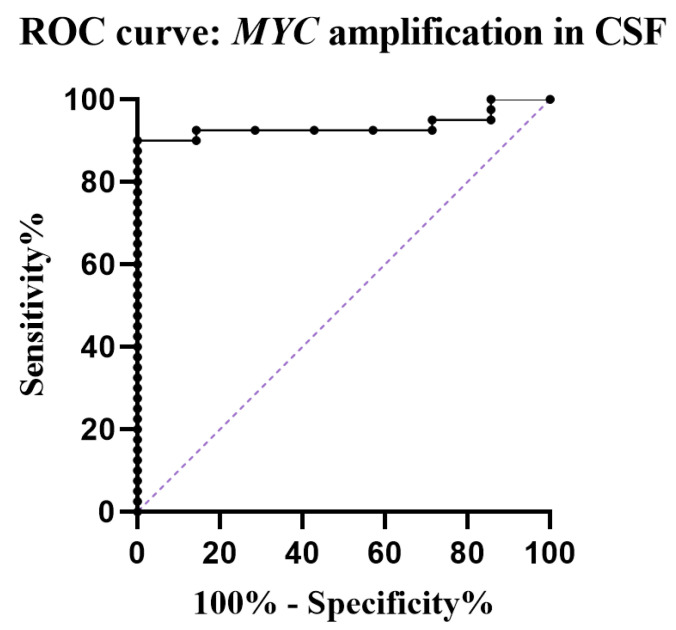
ROC curve showing sensitivity and specificity of ddPCR for detection of *MYC* amplification (AUC 0.93). With an optimal cut-off level of 2.2, the calculated sensitivity and specificity was 89.8% and 100%, respectively. Purple dotted line = reference line.

**Figure 3 cancers-15-02525-f003:**
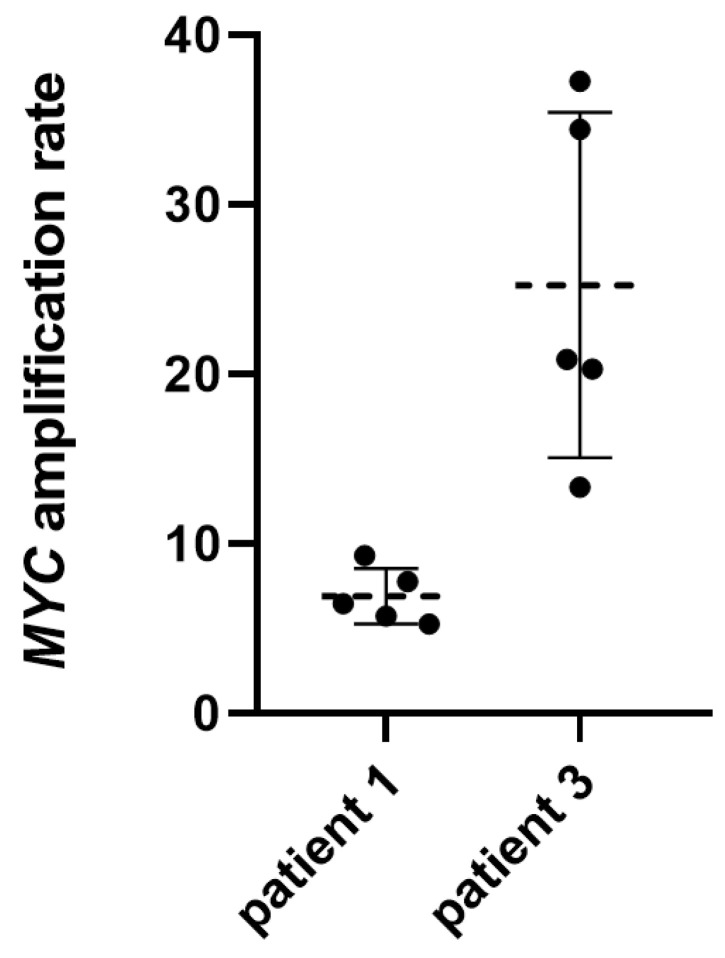
Results of ddPCR performed on five consecutive CSF samples from two patients. Black dots represent the mean of duplicate analyses of individual samples. Dotted lines represent the mean and whiskers the interquartile range.

**Figure 4 cancers-15-02525-f004:**
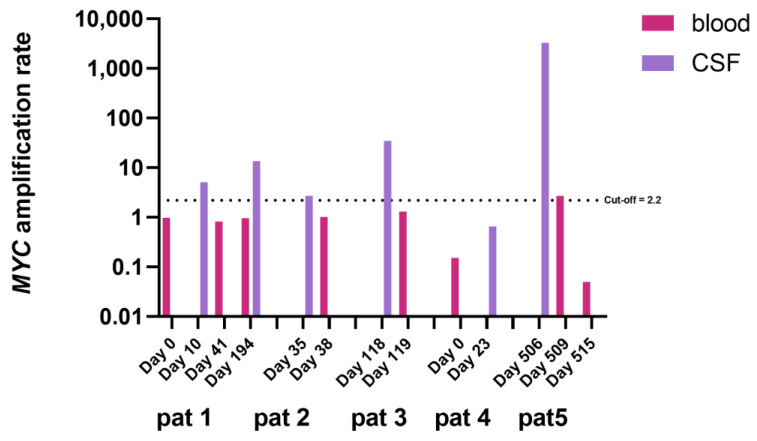
*MYC* AR in blood samples and matched CSF samples. Samples were matched by temporal proximity. Days marked at the x-axis represent the collection day of each sample starting at the day of first liquid biopsy for each individual patient. Note the log-axis at the left. Cut-off of 2.2 as depicted by the horizontal, dotted, black line.

**Figure 5 cancers-15-02525-f005:**
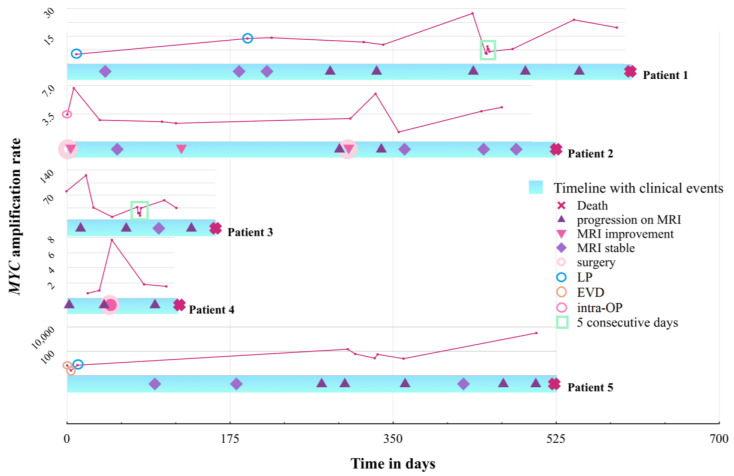
Swimmer plot depicting the timeline (time in days) of each patient starting at the date of first liquid biopsy until death. Symbols on the timelines are explained in the figure legend. MRI post-surgery was classified as improved if there was no progression at other sites. The results of ddPCR for *MYC* amplification detected in CSF are displayed for each patient individually just above their clinical timeline (in pink). LB sites are indicated by a circle within the AR diagram for each patient (see also Figure 6). The results of the sampling on 5 consecutive days are included in the line chart and marked by the turquoise square. Note the different scales for each patient given on the left. Scale for patient 5 is logarithmic.

**Figure 6 cancers-15-02525-f006:**
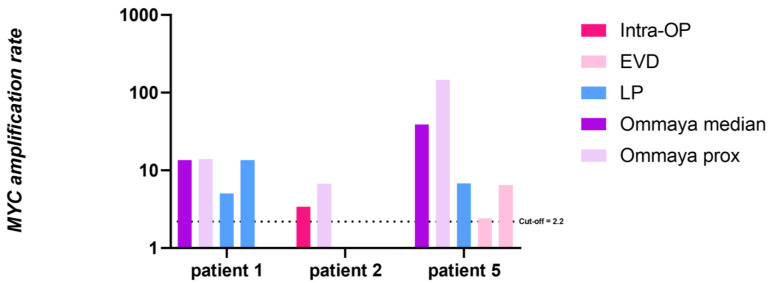
Comparison of *MYC* AR in CSF of patient 1, 2 and 5 from different sampling sites. For patient 1 the interval between 1st LP and 2nd LP was six months and 1 month from 2nd LP to 1st Ommaya puncture. For patient 2 the interval from intraoperative to 1st Ommaya puncture was 7 days and for patient 5 the EVD samples were from the 1st post-operative week, the LP was performed on day 14 and 1st Ommaya sampling was 10 months after initial surgery. LP: lumbar puncture; Intra-OP: intraoperative; Ommaya prox: 1st Ommaya sampling after sampling from another site. Ommaya median: median AR of all following Ommaya sampling time points. EVD: external ventricular drain. Cut-off at 2.2 AR marked by the black dotted line. Note the log scale for the y-axis.

**Table 1 cancers-15-02525-t001:** Comparison of *MYC* amplification rates in tumor tissue and CSF by different methods. For FISH analysis gain is defined as (ratio ≤ 4); OCCRA: Oncomine Childhood Cancer Research Assay (OCCRA) performed on IonTorrent (ThermoFisher Scientific, Waltham, MA, USA); TSO: True Sight Oncology 500 was performed on Illumina NextSeq Equipment (Illumina, Isc; San Diego, CA, USA); FISH: Fluorescense in situ hybridisation; for comparability reasons levels of *MYC* in CNV plots (Appendix A) were estimated and presented in the last column of Table 1. AR: amplification rate, AR calculation for ddPCR results = [(copies of gene of interest/copies of reference gene)] × 2; AR calculation from log2 copy number ratio (CNR) estimation: (2^CNR^)*2.

	**AR Tumor Tissue (Method)**	**AR Tumor Tissue by ddPCR**	**Median AR CSF**	**AR Estimation by CNV Plot Analysis**
Patient 1	not performed	17.2	9.7	3.0
Patient 2	gain in 27% amplification in 70% (FISH), mainly double minutes	no tumor tissue available	3.2	3.0
Patient 3	Amplification in 100%	69.9	34.8	4.3
Patient 4	gain in 64% of cells Amplification in 0% (FISH)	3.3	1.6	2.5
Patient 5	Amplification (FISH) AR of 40 (OCCRA) AR of 47 (TSO500)	39.2	25.9	3.0

## Data Availability

The data presented in this study is contained within the presented figures and Appendix A. Any additional data is available on request from the corresponding author, due to privacy restrictions.

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
