# Peer review of "Proof-of-Concept for Liquid Biopsy Disease Monitoring of MYC-Amplified Group 3 Medulloblastoma by Droplet Digital PCR"

_cancers, 2023, doi:10.3390/cancers15092525_

Round 1

Reviewer 1 Report

On the whole an interesting idea and study but data is preliminary and needs to developed with a larger multicenter study 

Author Response

We thank the reviewer for the important corrections and suggestions. We implemented all the corrections, as marked in red within the re-submitted document. 

The reviewer highlighted correctly, that CSF analysis in most cases was performed years later for method implementation and the illustrative case is used to demonstrate the potential importance of the method. We now clarified this in the materials and methods section, line 207-209. We further clarified it in the detailed case description “The fact that we detected MYC amplification in the CSF in our retrospective analysis, suggests that a tumor could have been suspected already upon analysis of CSF via lumbar puncture performed as diagnostic workup upon suspicion of optic neuritis.”, line 447-449.

We thank the reviewer for highlighting the limited readability of the swimmer plot (fig 5). We increased the font-size and adapted the colors. However, there is a lot of information contained in this graphic, so we would ask the editor for presentation within the paper as a large-scale graphic. We are happy to provide a high-resolution image. We also clarified the meaning of the swimmer plot in lines 357-366.

Reviewer 2 Report

The study employs ddPCR as a highly sensitive and specific method for detecting MYC amplification in the CSF of patients with medulloblastoma group 3. While the study is well-designed, a major limitation is the small sample size of patients included.

General comments:

The authors propose using Ommaya reservoirs, which are devices used for intraventricular therapy, as a method of obtaining CSF for patients with medulloblastoma group 3.

It would be appropriate to discuss the potential drawbacks of using these devices, as well as the proportion of patients with medulloblastoma group 3 who receive Ommaya reservoirs as part of their treatment and are subsequently eligible for ddPCR analysis with this approach.

Introduction:

The authors wrote: “Within our study, we highlight that the defining molecular alteration is better detectable in CSF than in biopsy tissue.”

The intended meaning of the authors' statement is ambiguous. It remains uncertain whether they propose defining molecular alterations in CSF is superior to defining them for example in tumor tissue. Clarification is needed.

Materials and Methods section

To improve the information presented, the authors may consider providing a brief description of the clinical and pathological characteristics of the patients and their tumors, including the WHO classification that the tumors follow.

The authors wrote: “Patients included in the study were treated for a CNS malignancy at the Department of Pediatrics and Adolescent Medicine, Medical University of Vienna. Results of methylation arrays from patients with MB were screened for the presence of MYC amplification.”

It is not clear if methylation arrays was done in tumor tissue DNA or on CSF ctDNA

Given the significance of methylation array results in patient selection for ddPCR analysis, it is advisable to provide a comprehensive description of the methylation classification method used, including the type of arrays utilized and the criteria for tumor classification (classifier). In addition, the methylation results of the specific samples should be mentioned in a reference or the copy number profiles of the tumors displaying MYC amplification following methylation array analysis should be included in a supplementary figure.

The authors wrote: “Except for minor modifications already established in our laboratory, such as the double elution with 20μL, the isolation of cf/ctDNA was performed according to the manufacturer’s instructions.”

It would be beneficial for the readers to provide further details on the methodology employed, especially for the double elution, including whether both fractions are mixed or analyzed independently. Additionally, please provide a reference for the modified method.

Results section

In the results section, it says: “One tissue sample from a group 3/4 MB patient with MYC was used as a control, showing no significant MYC amplification as per ddPCR.”

It is not clear how a control with MYC amplification did not show MYC amplification as per ddPCR. If this is not the case please clarify this sentence. Is this a positive or a negative control? Has the sample MYC amplification or not?

In Table 1 please explain how median AR in CSF is calculated? At which timepoint? Could it be possible to include an estimation of the level of amplification of MYC detected by methylation arrays in table 1?

In the manuscript text the authors talk about mean AR but they used median AR in table 1, why is this? Please clarify.

Suppl. Fig. 2 shows MYC amplification rate by ddPCR in the CSF of two patients (1&3) on five consecutive days

In this figure, the MYC AR varies considerably for patient 1, suggesting that this method may not be reproducible. It would be helpful if the authors could comment on the reasons for these fluctuations and how they could affect the reliability of the method on detecting disease progression.

Author Response

The study employs ddPCR as a highly sensitive and specific method for detecting MYC amplification in the CSF of patients with medulloblastoma group 3. While the study is well-designed, a major limitation is the small sample size of patients included.

We thank the reviewer for their important comments and suggestions that we happily incorporated in our manuscript:

We are aware of the limited patient number and hope that our manuscript raises awareness on the importance of liquid biopsy sampling within the pediatric neurooncology community in order to make lager studies feasible.

General comments:

The authors propose using Ommaya reservoirs, which are devices used for intraventricular therapy, as a method of obtaining CSF for patients with medulloblastoma group 3.

It would be appropriate to discuss the potential drawbacks of using these devices, as well as the proportion of patients with medulloblastoma group 3 who receive Ommaya reservoirs as part of their treatment and are subsequently eligible for ddPCR analysis with this approach.

We thank the reviewer for this important remark on the impact of our study. Intraventricular therapy is part of the first-line therapy of infants suffering from MB but not eligible for radiotherapy (e.g. HIT-MED guidance) and is frequently applied in recurrent MBs as part of diverse regimes (e.g. MEMMAT). Especially in patients with highly aggressive tumors such as MYC amplified MB intraventricular therapy is an important part of therapy, frequently leading to implantation of Ommaya or Rickham reservoirs --> added in line 139-146; the potential risks and their frequency are mentioned and referred to our previous study (highlighted in yellow (137-139)).

However, as a comment, we would not recommend the implantation of an Ommaya reservoir for the sole purpose of liquid biopsy based on our current knowledge.

Introduction:

The authors wrote: “Within our study, we highlight that the defining molecular alteration is better detectable in CSF than in biopsy tissue.”

The intended meaning of the authors' statement is ambiguous. It remains uncertain whether they propose defining molecular alterations in CSF is superior to defining them for example in tumor tissue. Clarification is needed.

We thank the referee for this comment. The intended meaning was to highlight the potential of ddPCR based LB as an additional highly sensitive tool in our previous study (Baumgartner, Stepien et al.), now clarified in line 152-164.

“Within this study, we found that the defining molecular alteration was better detectable in CSF than in the biopsy tissue, highlighting the sensitivity of this method. Therefore, LB might also aid in the timely diagnosis of unspecific CNS lesions.”

We further clarified the main points of our study at the end of the introduction. Line 167-173

In the present study, we selected ddPCR as detection method of MYC amplification, utilizing its extremely high sensitivity for singular molecular alterations. We show that this molecular alteration can be tracked in CSF of MB patients. Importantly, liquid biopsy detection of MYC amplification harbors potential in disease staging and monitoring as well as early diagnosis as further discussed in our representative case.

Materials and Methods section

To improve the information presented, the authors may consider providing a brief description of the clinical and pathological characteristics of the patients and their tumors, including the WHO classification that the tumors follow. 

The authors wrote: “Patients included in the study were treated for a CNS malignancy at the Department of Pediatrics and Adolescent Medicine, Medical University of Vienna. Results of methylation arrays from patients with MB were screened for the presence of MYC amplification.”

It is not clear if methylation arrays was done in tumor tissue DNA or on CSF ctDNA

Given the significance of methylation array results in patient selection for ddPCR analysis, it is advisable to provide a comprehensive description of the methylation classification method used, including the type of arrays utilized and the criteria for tumor classification (classifier). In addition, the methylation results of the specific samples should be mentioned in a reference or the copy number profiles of the tumors displaying MYC amplification following methylation array analysis should be included in a supplementary figure.

This is a valuable suggestion by the reviewer! For clarification, we added a supplementary table including patients characteristics, details on histology and on methylation based classification. Referred to in line 182-183 and added as supplementary table 1; leading to a change in the naming of the other supplementary tables.

The Methylation array was performed on tumor tissue, which is now clarified in line 184. We also specified the Methylation analysis methods used (Table S1) and added CNV plots of the MYC locus used for estimation of MYC amplification as well as scatter plots of the ddPCR results as Figure S1. See line 185-189. (based on log2 values outlined in the CNV plots)

The authors wrote: “Except for minor modifications already established in our laboratory, such as the double elution with 20μL, the isolation of cf/ctDNA was performed according to the manufacturer’s instructions.”

It would be beneficial for the readers to provide further details on the methodology employed, especially for the double elution, including whether both fractions are mixed or analyzed independently. Additionally, please provide a reference for the modified method.

We do apologize if this was not outlined clearly in the manuscript. There was only one modification to the manufacturers’ instructions which was using the same column with the same eluate a second time in order to increase the concentration of eluted cfDNA. We routinely apply this method in our laboratory, however, since it is a minor modification we have not published it separately in a manuscript. To answer the reviewers comment we clarified the method description. See 240-242.

Results section

In the results section, it says: “One tissue sample from a group 3/4 MB patient with MYC was used as a control, showing no significant MYC amplification as per ddPCR.”

It is not clear how a control with MYC amplification did not show MYC amplification as per ddPCR. If this is not the case please clarify this sentence. Is this a positive or a negative control? Has the sample MYC amplification or not?

Thank the reviewer for this important comment and do apologize for this editing error. Obviously, we accidentally erased a word during manuscript editing, it was a negative control with MYC polysomy, corrected in line 286.

“One tissue sample from a group 3/4 MB patient with MYC polysomy was used as a control”

In Table 1 please explain how median AR in CSF is calculated? At which timepoint? Could it be possible to include an estimation of the level of amplification of MYC detected by methylation arrays in table 1?

We thank the reviewer for this suggestion to outline the AR rate more clearly. The different methods for calculation of AR are defined in the materials and method section, line 254, highlighted in yellow. Additionally, we added the formula again in the comments for table 1 (299-301, highlighted in yellow). We included an estimation of the amplification level of MYC derived from the CNV plots in table 1, and thank for this suggestion.

In the manuscript text the authors talk about mean AR but they used median AR in table 1, why is this? Please clarify.

This is a highly important remark which we hope to clarify. In the course of our analyses we initially compared mean values of patients. However, given the high level of MYC in some cases in the course of disease and thus variability in the longitudinal measurements we applied median in the end. We now uniformly use the median for comparison of patients and description of the cohort, which we implemented now throughout the manuscript (highlighted in yellow). The only case where we sticked to the mean was for investigating variability of 5 consecutive liquid biopsy samples within one week (figure 3) as we deemed this more appropriate in this case (see next comment).

Suppl. Fig. 2 shows MYC amplification rate by ddPCR in the CSF of two patients (1&3) on five consecutive days

In this figure, the MYC AR varies considerably for patient 1, suggesting that this method may not be reproducible. It would be helpful if the authors could comment on the reasons for these fluctuations and how they could affect the reliability of the method on detecting disease progression.

We thank the reviewer for pointing out this interesting finding. The extent of variation is an important part of our results and is of particular interest when comparing different time spans. Whereas the variability within one week was rather small (patient 1 5.3-9.3; patient 3 13.3-37.3) it was considerably larger when looking at the whole course of disease (patient 1 5.0-27.2, patient 3 9.9-125.9). This is also outlined in figure 5 (Swimmer plot). We highlighted this in lines 361-364 by additionally including the overall range of measured values. Nevertheless, we do fully agree that it is important to keep in mind the variation when interpreting results. We perceive the main benefit of our method in the detection (negative/positive), and clinical correlation when it comes to important clinical changes; to determine the importance of small variations higher patient and sample numbers are needed. We highlighted our previous considerations in yellow (524-529) and elaborated further on this issue in lines 530-543.